



# Importance of the advection scheme for the simulation of water isotopes over Antarctica by general circulation models: a case study with LMDZ-iso (LMDZ5a revision 1750)

Alexandre Cauquoin[1], Camille Risi[2]

[1]Alfred Wegener Institute, Helmholtz Centre for Polar and Marine Sciences, Bremerhaven, Germany
[2]Laboratoire de Météorologie Dynamique/Institut Pierre-Simon Laplace (LMD/IPSL), CNRS, Sorbonne Universités, UPMC Univ Paris 06, Paris, France

*Correspondence to*: Alexandre Cauquoin (alexandre.cauquoin@awi.de)

**Abstract.** Atmospheric general circulation models (AGCMs) are known to have a warm and isotopically enriched bias over Antarctica. We test here the hypothesis that these biases are consequences of a too diffusive advection. Using the LMDZ-iso model, we show that a good representation of the advection, especially on the horizontal, is very important to reduce the bias in the isotopic contents of precipitation above this area and to improve the modelled water isotopes – temperature relationship. A good advection scheme is thus essential when using GCMs for paleoclimate applications based on polar water isotopes.

## 1 Introduction

Water stable isotopes ($H_2^{16}O$, HDO, $H_2^{18}O$, and $H_2^{17}O$) are integrated tracers of the water cycle. Especially, the isotopic composition recorded in polar ice cores enabled the reconstruction of past temperature variations (Jouzel, 2013 and references therein). For example, low accumulation sites in the central Antarctic plateau provided the longest ice core records, allowing to reconstruct past climate over several glacial-interglacial cycles (Jouzel et al., 2007). However, the interpretation of isotope signals remains challenging because of the numerous and complex climate processes involved, especially in this part of the world subject to extreme weather conditions.

To improve our knowledge on the mechanisms controlling the water isotopes distribution, atmospheric general circulation models (AGCMs) enhanced by the capability to explicitly simulate the hydrological cycle of the water isotopes are now frequently used (Joussaume et al., 1984; Risi et al., 2010a; Werner et al., 2011). Water isotopes in climate models have been used, for example, to better understand how the climatic signal is recorded by isotopes in polar ice cores at paleoclimatic time scales (Werner et al., 2001).



However, some issues remain concerning the simulation of the climate over the Antarctic continent by AGCMs. Indeed, they frequently present a warm bias over this area (Masson-Delmotte et al., 2006) and isotopic values in precipitation that are not depleted enough compared to observations (Lee et al., 2007; Risi et al., 2010a; Werner et al., 2011). This raises the question why many of the AGCMs have these warm and enriched in heavy water isotopes biases over Antarctica.

In this paper, we hypothesize that these biases are associated with an excessively diffusive water vapor transport, i.e. transport that is associated with too much mixing. Dehydration of air masses by mixing with a drier air mass leads to more enriched water vapor than dehydration by condensation and associated Rayleigh distillation (Galewsky and Hurley, 2010). For the same reason, poleward water vapor transport by eddies (which act as mixing) leads to more enriched water vapor in

Antarctica than transport by steady advection (Hendricks et al., 2000). In AGCMs, the simulation of humidity and of its water isotope contents is sensitive to the diffusive property of advection schemes. For example, the excessive diffusion during vertical water vapor transport seems to be the cause of the moist bias found in most AGCMs in the tropical and subtropical mid and upper troposphere, and of the poor simulation of isotopic seasonality in the subtropics (Risi et al., 2012). The diffusivity of the advection scheme in the vertical has also important consequences on modeling of tracers like tritium,

which is related to stratospheric moisture input (Cauquoin et al., 2016).

The goal of this paper is to test if the warm and enriched biases in Antarctica are associated with an excessively diffusive water vapor transport. This paper follows the work of Cauquoin et al. (2016) who compare two advection schemes, which differ by their diffusive character.


## 2 Model, simulations and method

We use here the isotopic AGCM LMDZ-iso (Risi et al., 2010a) following the model setup from Cauquoin et al. (2016) at a standard latitude-longitude grid resolution of 2.5° × 3.75°, and with 39 layers in the vertical spread in a way to ensure a realistic description of the stratosphere and of the Brewer-Dobson circulation (Lott et al., 2005). The model has been

validated in polar regions for the simulation of both atmospheric (Hourdin et al., 2006) and isotopic (Risi et al., 2010a) variables.

To quantify the effects of the prescribed advection scheme on water stable isotope values over Antarctica, we performed four sensitivity simulations: (1) one control simulation with the van Leer (1977) advection scheme (called VL), which is a second

order monotonic finite volume scheme prescribed in the standard version of the model (Risi et al., 2010a); and three other simulations whose the van Leer advection scheme has been replaced by a single upstream scheme (Godunov, 1959), intrinsically more diffusive as explained below, (2) on the horizontal plane (UP_xy), (3) on the vertical direction (UP_z), and (4) on every directions (UP_xyz). Indeed, depending on one tunable parameter, the LMDZ model can be used with these 2





versions of the advection scheme according to the object of study (Risi et al., 2012). The advection scheme in the simulations presented the LMDZ-iso reference paper from Risi et al. (2010a) was set erroneously as the simple upstream scheme rather than van Leer's scheme (Risi et al., 2010b), and has little influence on their results.

In the sake of simplicity, we consider the advection along one dimension only, with wind flowing from grid box $i$-1 to grid box $i$ and from grid box $i$ to grid box $i$+1. In both Van Leer's (1977) second-order advection scheme and the upstream advection scheme (Godunov, 1959), the mixing ratio after advection in box $i$ ($q_i'$) is given by:

$$q_i' = \frac{q_i \times m_i + U_{i-1/2} \times q_{i-1/2} - U_{i+1/2} \times q_{i+1/2}}{m_i + U_{i-1/2} - U_{i+1/2}}, \qquad (1)$$

where $q_i$ and $m_i$ are the mixing ratio and air mass in box $i$, $U_{i-1/2}$ is the air mass flux at the boundary between boxes $i$ and $i$-1,
$U_{i+1/2}$ is the air mass flux at the boundary between boxes $i$ and $i$+1. The two schemes differ in the way the water vapor mixing ratio that is advected from box $i$-1 to $i$, $q_{i-1/2}$, and the water vapor mixing ratio that is advected from box $i$ to $i$+1, $q_{i+1/2}$, are calculated.

In Van Leer's scheme, $q_{i-1/2}$ is a linear combination of the mixing ratio in the boxes $i$-1 and $i$. Similarly, $q_{i+1/2}$ is a linear combination of the mixing ratio in the boxes $i$ and $i+1$. For example, if the air mass flux from grid box $i$-1 to grid box $i$ is
very small, then $q_{i-1/2} = (q_i + q_{i-1})/2$. This reflects the air that is advected into box $i$ is restricted to a small margin along the $i$-1/$i$ boundary, so its mixing ratio is exactly intermediate between $q_{i-1}$ and $q_i$.

In contrast, the upstream scheme is much simpler: $q_{i-1/2} = q_{i-1}$ and $q_{i+1/2} = q_i$. This means that even if the air mass flux from grid box $i$-1 to grid box $i$ is very small, the air that is advected into box $i$ has the same water vapor mixing ratio as grid box $i$-1 as a whole. This makes the upstream scheme much more diffusive.


We use here the LMDZ-iso outputs for the period 1990-2008. We express the isotopic composition of difference water bodies in the usual δ-notation as the deviation from the Vienna Standard Mean Ocean Water (V-SMOW). So for $H_2{}^{18}O$, the δ$^{18}$O value is calculated as δ$^{18}$O = ([$H_2{}^{18}O$]/[$H_2{}^{16}O$])$_{sample}$ / ([$H_2{}^{18}O$]/[$H_2{}^{16}O$])$_{V\text{-}SMOW}$ − 1) × 1000. Long-time mean δ values are then calculated as precipitation-weighted mean. For the quantitative model-data comparisons, we interpolate the LMDZ-
iso simulation outputs from the model grid to the same geographical coordinates as the observational datasets. We make use of the observational database compiled by Masson-Delmotte et al. (2008) for analyzing the model performance over Antarctica. We also focus especially on the East-Antarctic plateau (defined by the black bold contour of 2500 m above sea level elevation in Figure 1) because this essential area provides us the main reconstructions of past climate and that it constitutes an extreme test for isotope-enabled AGCMs.




## 3 Results and discussion

Figure 1 shows the observed annual mean $\delta^{18}O$ values in the snow surface in Antarctica compiled by Masson-Delmotte et al. (2008) (Figure 1a) and the difference with the modeled annual $\delta^{18}O$ in precipitation from the UP_z (Figure 1b), UP_xy (Figure 1c) and VL (Figure 1d) simulations. Our simulated $\delta^{18}O$ in precipitation is very sensitive to the choice of the

advection scheme, with more enriched values when a more diffusive advection scheme is applied. The results from the VL simulation are in better agreement with the $\delta^{18}O$ observations over Antarctica (Figure 1d). This is confirmed by the root-mean-squared errors (RMSE) of simulated $\delta^{18}O$ in precipitation from the UP_xyz and VL simulations, calculated as the difference between the observed annual mean values and the LMDZ-iso results, which are of 7.97‰ and 4.47‰ respectively (i.e. 21.7% and 12.2% of the observed mean Antarctic $\delta^{18}O$ value). The results from the VL simulation for the considered

other variables, δD and annual mean surface temperature, are the closest of the observations with RMSE of 40.93‰ and 6.60°C respectively (Table 1, red background). According to the observations, the East-Antarctic plateau is where the water isotope values are the lowest (mean $\delta^{18}O$ below -40‰, Figure 1a) due to the very low temperatures taking place. Because of the extreme climate conditions at this area, one can see that the main disagreements between model outputs and observations are located at this place (Figure 1 and blue background of Table 1). Again, the outputs from the VL simulation are in better

agreement with the observations (Table 1, blue background). These first results confirm that an excessively diffusive water vapor transport influences significantly the simulated isotopic and temperature values over Antarctica.

The bias in temperature is deteriorated about in the same way when applying a more diffusive advection on the vertical direction or on the horizontal plan, as shown with the RMSE of annual mean temperature of 7.50, 7.31 and 6.60°C for the

UP_z, UP_xy and VL simulations respectively (Table 1, red background). This tendency is the same when focusing on the East-Antarctic plateau. To explain the influence of the advection on the temperature over Antarctica, one can hypothesize that the Antarctic continent is better isolated, and so colder, when the advection of the model is less diffusive. It has also been suggested that the Antarctic warm bias in AGCMs could be linked to the general poor representation of the polar atmospheric boundary layer and related atmospheric inversion temperatures in these models (Krinner et al., 1997). A too

diffusive advection could accentuate the misrepresentation of this boundary layer. However, the average temperatures of -29.56°C and -31.54°C from the UP_xyz and VL simulations compared the mean observed temperature (-36.93°C) shows that the excessive diffusion of the advection scheme explains only 27% of the Antarctic total warm bias. Its main cause is beyond the scope of our study.

We now compare the model-data difference in $\delta^{18}O$ for the different simulations. The UP_z simulation (upstream vertical advection, Figure 1b) increases a little the bias in $\delta^{18}O$, but its results stay relatively close of the $\delta^{18}O$ values from the VL simulation, indicated by the similar average values of the model-data difference for all Antarctica of 1.91‰ and 1.02‰ with the UP_z and VL simulations respectively. On the other hand, the $\delta^{18}O$ outputs from the UP_xy simulation (upstream





horizontal advection, Figure 1c) display greater differences with the isotopic data, as revealed by the bigger mean model-data difference in $\delta^{18}O$ of 5.33‰. This is even more significant when focusing on the East-Antarctic plateau, with a model-data difference in $\delta^{18}O$ reaching 20‰ at some locations. The RMSE of simulated $\delta^{18}O$ and $\delta D$ values as compared to the observations amount to 9.69‰ and 76.44‰ with the UP_xy simulation and to 3.80‰ and 33.58‰ with the VL simulation

(Table 1, blue background). All our results show that the too diffusive water vapor transport through the advection scheme in LMDZ, especially on the horizontal plan, contributes for 82% and 76% to the enriched bias in $\delta^{18}O$ and $\delta D$ of precipitation over Antarctica. The fact that the advection scheme affects substantially the simulated $\delta^{18}O$ but only marginally temperature suggests that the impact on $\delta^{18}O$ is not through temperature. This lends support to our hypothesis that too much diffusive mixing along the poleward transport leads to overestimated $\delta^{18}O$ because dehydration by mixing follows a more enriched

path than dehydration by Rayleigh distillation (Hendricks et al., 2000; Galewsky and Hurley, 2010).

We compare now our simulated spatial $\delta^{18}O$–temperature relation and $\delta^{18}O$ values for a given temperature to the ones from the data compiled by Masson-Delmotte et al. (2008). The spatial gradient for all the dataset is at 0.80‰/°C. The same gradient according to the UP_z, UP_xy and VL simulations is of 0.79, 0.69 and 0.83‰/°C. It confirms that the diffusive

property of the advection scheme on the horizontal plan is essential to better model the water isotope distribution over Antarctica. We make the same comparison but by restricting the dataset to the ones on the East-Antarctic plateau (Figure 2). As noticed previously, the modeled temperature is globally overestimated whatever the simulation considered. Especially, no simulated temperature reaches a value below -50°C, and our $\delta^{18}O$ values are too depleted for the lowest simulated temperature range between -49°C and -45°C. The good agreement between the simulated isotopic values in precipitation

over the East-Antarctic plateau (Figure 1d) and the observations despite too warm simulated temperatures could be explained by a bad representation of the atmospheric boundary layer and of its related inversion temperature. This would give a too warm surface temperature even if the condensation temperature in the model was close of the reality. This has the consequence that the spatial $\delta^{18}O$–temperature gradients on the East-Antarctic plateau deduced from the models (thin colored lines in Figure 2) are too steep compared to the observed one (0.85‰/°C, black line). If we restrict the fit to the modeled

temperatures over -40°C, corresponding to the change of slope in our modeled $\delta^{18}O$–temperature relations (thick colored lines), the obtained gradients are then in much better agreement with the one from the observations. In addition to have water isotope and temperature values in better agreement with the data, the VL simulation exhibits the closest $\delta^{18}O$–temperature gradient (0.87‰/°C) of the one from the observations. Finally, for a given temperature value, our modeled $\delta^{18}O$ values are more enriched when the advection is more diffusive especially on the horizontal plan (red crosses on Figure 2). This is

consistent with our hypothesized mechanism for the impact of the advection scheme on simulated $\delta^{18}O$, which is based on the more enriched $\delta^{18}O$ path followed by air masses when dehydrated by mixing compared to when dehydrated by Rayleigh distillation.




## 4 Conclusions

We have tested with LMDZ-iso if the warm and isotopically enriched biases in Antarctica, frequently observed in the AGCMs, are associated with the diffusive property of the advection scheme. The excessive diffusion can explain a part of the Antarctic temperature bias, but is not the main cause. The simulated water isotope contents in Antarctica are very

sensitive to the diffusive character of the water vapor transport, especially on the horizontal plan. More the contribution of mixing (i.e. diffusion) is important, more precipitation is enriched in heavy isotopes. So we conclude here that the excessive numerical diffusion has a large influence on the enriched bias and the isotope–temperature relation over Antarctica observed in LMDZ-iso. These findings are even more important for the East-Antarctic plateau where are the main ice cores allowing paleoclimate reconstructions. This demonstrates that one must be careful that an AGCM has a good representation of the

advection scheme, especially on the horizontal domain, to simulate water isotopes over Antarctica. The next version of LMDZ, LMDZ6, should improve this aspect with a higher spatial resolution.

## 5 Code availability

The code of the LMDZ5a model (revision 1750 without water isotopes), on which LMDZ-iso is based on, can be downloaded via the command: svn checkout http://svn.lmd.jussieu.fr/LMDZ/LMDZ5/branches/testing@1750 LMDZ5.

General information about the model and its documentation can be found on http://lmdz.lmd.jussieu.fr and on http://lmdz.lmd.jussieu.fr/utilisateurs/manuel-de-reference-1/lmdz5-documentation/view respectively. The LMDZ-iso code is available as a supplement of this manuscript.

*Acknowledgements*. We thank A. Landais for her useful suggestions on this manuscript. LMDZ simulations have been

performed on the Ada machine at the IDRIS computing center under the GENCI project 0292. The research leading to these results has received funding from the European Research Council under the European Union's Seventh Framework Programme (FP7/20072013)/ERC grant agreement no. 30604.

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



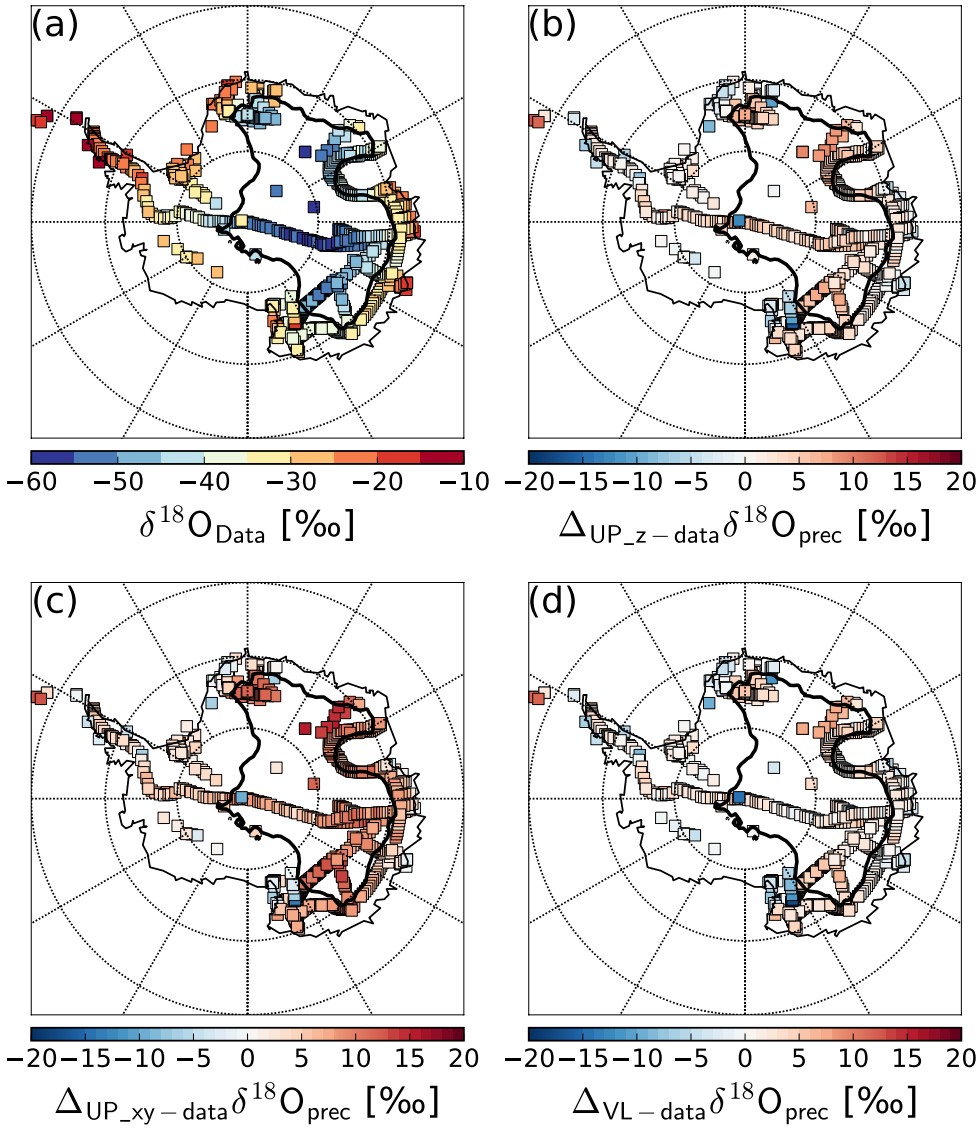

**Figure 1: Map of Antarctica showing (a) the observed δ$^{18}$O values from the compilation by Masson-Delmotte et al. (2008), (b) the difference between the simulated δ$^{18}$O in precipitation and the δ$^{18}$O observations for the UP_z, (c) UP_xy and (d) VL simulations. The bold black line shows the contour of 2500 m above sea level elevation.**




| | Mean data | Mean UP_z | RMSE UP_z | Mean UP_xy | RMSE UP_xy | Mean VL | RMSE VL |
|---|---|---|---|---|---|---|---|
| T (°C) | **-36.93** | -30.51 | 7.50 | -30.69 | 7.31 | -31.54 | 6.60 |
| $\delta^{18}$O (‰) | **-36.76** | -34.85 | 4.84 | -31.43 | 7.63 | -35.74 | 4.47 |
| $\delta$D (‰) | **-289.62** | -272.28 | 43.76 | -251.34 | 62.00 | -279.49 | 40.93 |
| T (°C) | **-47.46** | -39.49 | 8.38 | -39.88 | 7.99 | -40.71 | 7.23 |
| $\delta^{18}$O (‰) | **-46.77** | -42.27 | 5.03 | -37.37 | 9.69 | -43.76 | 3.80 |
| $\delta$D (‰) | **-366.98** | -325.37 | 43.79 | -291.99 | 76.44 | -336.25 | 33.58 |

Table 1: Mean and RMSE of annual mean distributions of observed and simulated temperature (T), $\delta^{18}$O and $\delta$D for the full Antarctic dataset (red background) and restricted to the East-Antarctic plateau (blue background).

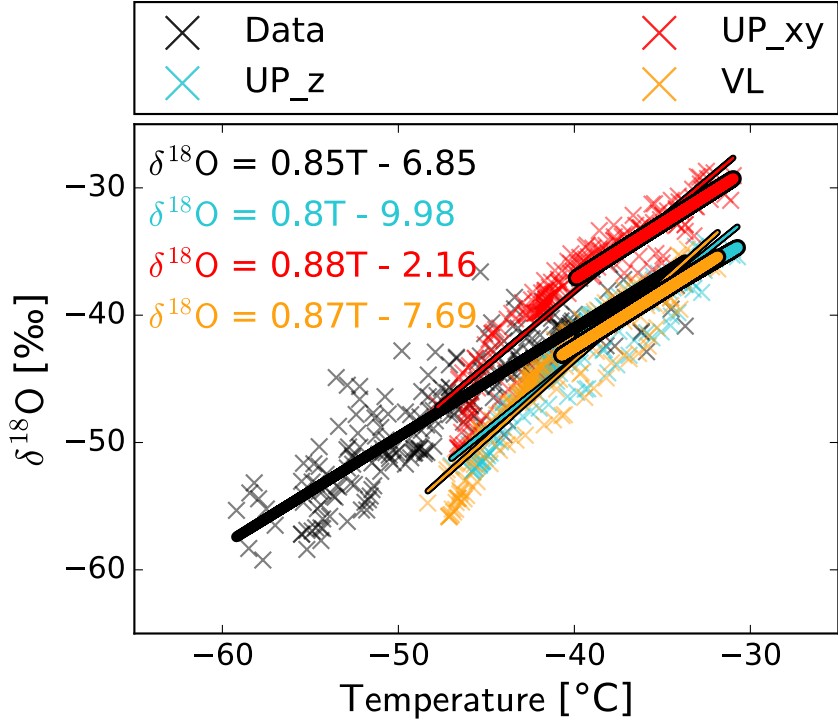

Figure 2: Relationship between $\delta^{18}$O and temperature on the East-Antarctic plateau according to the observations (black) and the UP_z (blue), UP_xy (red) and VL (orange) simulations. For each simulation outputs, two linear regressions have been conducted: one on the full East-Antarctic plateau dataset (thin lines) and one on the same dataset without the temperatures below -40°C (bold lines). The corresponding formulas of these latters are also shown.