# Peer review of "Importance of the advection scheme for the simulation of water isotopes over Antarctica by general circulation models: a case study with LMDZ-iso (LMDZ5a revision 1750)"

_Geoscientific Model Development, 2017_

## Referee Comment (RC1) · Anonymous Referee #1 · 21 Aug 2017

In their paper, A. Cauquoin and C. Risi apply two different advection schemes (one of which in three different versions) in a GCM to test the influence of the representation of the advection on temperature and water isotope ratios in Antarctic precipitation. The paper is well structured and provides some interesting and important points on the uncertainties of Antarctic climate reconstruction that stems from model advection. In part, the paper lacks depth in method and interpretation of the results. The main conclusion that a physically more advanced advection scheme leads to more realistic

results than a relatively simple one is not striking, however, the analysis is still a worthy endeavour. Before publication in GMD, I suggest to consider two major points, one concerning the method and one concerning the discussion, and a couple of minor points that are listed below.

**Major comments:**

- The paper only compares two advection schemes, one of which is (as described on page 3) obviously physically not as advanced as the other one. Hence, as already stated above, it is not surprising that the one captures the observations better. Some points can still be gained through this comparison, but it is a little disappointing that there are no further comparisons made. I am not aware of how complicated it is to run LMDZ with other advection schemes, but the study could go a lot further through that (and apparently this had even already been done in Cauquoin et al (2016), but the conclusions of that study are not discussed here). At the end it could be discussed what sort of GCM advection scheme ((semi-)Lagrangian, finite differences, explicit Eulerian,...) can be recommended to represent these processes well. The only, rather poor statement of the paper is that not too diffusive is better.
Another form of enhancing the paper would be to try different horizontal resolutions. I assume that a higher horizontal resolution could, in particular for the "low-order" advection scheme, make a big impact in representing isotope ratios and temperature. Werner et al. 2011 show some study on this in Antarctica, here this issue is not even discussed (it only appears in the conclusion section without any connection to the rest of the paper). Some more work on this could easily enhance the (now pretty thin) message of the paper.

- The explanation for the conclusion that the temperature bias plays a minor role only for the isotope ratio bias is not fully convincing.
Kinetic isotope fractionation is highly non-linear, in particular during desublima-

tion of ice from the gas phase, which particularly happens at very low temperatures. A few degrees can have a large effect on the kinetic fractionation factor and thereby cause large differences in the isotope ratios. Fig. 2 also suggests a nonlinear relationship between temperature and $\delta^{18}$O below -40°. Your statement that the relations agree better above -40° strengthens the assumption that the temperature differences between model and observations are crucial here, also because the model does never simulate temperatures below -50°C. When you argue with the differences in RMSE here, you take this relation on to be purely linear, and that does not seem to hold for the entire range of data. Also, to my knowledge there are several microphysical isotope processes in the cloud and convection schemes not fully or not at all represented in the model water isotope physics. That should probably be discussed in this context, because it might have an impact at some temperature range.

Since this is an important point for the conclusions of the paper, I suggest to add some more analysis and discussion on this topic to provide a convincing and more nuanced argumentation.

**Minor comments:**

- The abstract lacks to make the point about the impact of the temperature bias on the isotope ratio bias. I think this is a crucial point, but see also my major point.

- P1 L11 and L13: "good" is a very relative term. This could also mean computationally efficient, and that would probably not match "your" meaning "good".

- P1L16: You know that these are isotop**ologues**. Please state this once and say that you call it isotopes (like in Werner et al., 2011). Is H$_2^{17}$O included in your model? Why don't you evaluate it also in the study?
  Also, this sentence leaves open if you are talking about the "real world" or your model. In the real world there are a couple more water isotopologues.
- P1L18: Please explain very briefly what low accumulation sites are.

- P1L20: Please name/list some of those "complex climate processes".

- P1L21: . This is particularly the case for the Antarctic, because this part of the world is subject to extreme weather conditions.

- P2L1: Change "Indeed" to "For example"

- P2L10-15: Here you talk only about studies that focus on vertical transport, although you said you would focus on horizontal advection, that is confusing. Is there no other work on horizontal advection with that respect?
  In the next paragraph you cite Cauquoin et al (2016), who compare two advection schemes. This seems to be relevant for the present paper, but you do not summarise their results in the context of the goal of the present study. Why not?

- P2L32: "intrinsically more diffusive as explained below". This statement dangles in the nowhere here and confuses a little. You should either dedicate an entire sentence to the point or wait with it until it is explained below.

- P2L33: Remove "indeed"

- P3L2-3: ...presented in the ...erroneously to the ... rather than to the van Leer scheme ...

- P3L3: What results are that? And why do you nevertheless expect that fact to have a considerable influence on **your** results then?

- P3L5 For the sake...

- P3L19: But also, it makes the upstream scheme physically not very sensible, right? Why would you use such an advection scheme anyway? Does it have advantages over the other scheme, or generally over other advection schemes? This is also part of my major point.

- P3L21: For reproducibility, can you say what SSTs you use? Is the simulation free running? Why this period? Is this the period where most data is available? Do you average the data over this period too?

- P3L24-25: Can you estimate the uncertainty that bears the resolution of your model and this interpolation?

- P3L28: Why is this area "essential"?

- P3L28: ...provides the main...climate and it ...

- P3L29: Why? Do other regions in Antarctica not constitute an extreme test?

- P4 and table 1: Why don't you include the results of the "xyz" simulation to the table as well? No need to include it to figure 1, but in the table it wouldn't harm.

- P4 end of second paragraph: You should restructure this paragraph, this way it sounds odd. Suggestion:
  ↦ Move the sentence about the poor representation of the boundary layer to the end of the paragraph.
  ↦ Add another process that could possibly explain the rest of the bias
  ↦ State that the investigation of these processes wrt the bias lies beyond the scope of this study.

- P5L15: and elsewhere too: horizontal plane

- P5L17: You do not really mean "globally", right? In all Antarctica?

- P5L11: The aspect about higher resolution was never discussed, now it pops up in the conclusions, that should not be. Anyway, this is also a part of my major point.

---

## Referee Comment (RC2) · Anonymous Referee #2 · 27 Oct 2017

Cauquoin and Risi have investigated the effect of the advection scheme on the simulation of the water isotopes and temperatures over Antarctica. By conducting sensitivity tests with the LMDZ-iso model, they have concluded that the diffusivity of the advection scheme on the horizontal plane is crucial to simulate the water isotopes and temperatures over Antarctica. The paper is well written and structured. However, the scientific finding is very thin. The unacceptable level of the diffusivity property of the upstream scheme is a well-known fact and the sensitivity results are not surprising.

Therefore, I cannot recommend this manuscript for publication at this stage. As the authors argued, the diffusive property of AGCMs is attributable to the performance of the advection scheme, but it is at a much higher level than that of the upstream scheme. I believe that the upstream scheme is no longer used in current state-of-the-art AGCMs. Even the van Leer scheme is known as a diffusive scheme and using a less diffusive scheme is a challenging task. Actually, there are a number of more sophisticated advection schemes, but most of them are computationally expensive. Therefore, one may consider that the van Leer scheme is a reasonable choice in terms of accuracy and computational cost. I suggest that the authors should try more sophisticated schemes than the van Leer one and compare results in terms of accuracy and computational cost. Presumably, the computational cost is an important issue for paleoclimate studies, because they require very long-term simulations. Hourdin and Armengaud (1999) might be a good reference paper, in which not only the upstream scheme and van Leer schemes, but also other sophisticated schemes were tested based on the LMDZ model. If such advection scheme tests are done for the water isotopes and discussion is made from the paleoclimate view, that would be an attractive paper.

Minor comments:

P1. L13: "good" is ambiguous.

P3, L1-L3: I cannot understand what this sentence means.

P5, L7-9: "The fact that the advection. . .is not through temperature" is not fully convincing.

P6, L2-3: "The excessive diffusion . . . but is not the main cause" should be removed from Conclusion, because "its main cause is beyond of the scope of the study" (P4, L28).

Reference:

Hourdin, F., and A. Armengaud, 1999: The use of finite-volume methods for atmospheric advection of trace species. Part I: Test of various formulations in a general circulation model. Mon. Wea. Rev., 127, 822–837.

---

## Author Comment (AC1) · 27 Nov 2017

**Response to the reviewers**

We acknowledge the two referees for their reviews and constructive comments that helped to improve this manuscript. We have revised it as described in detail below, and we hope that we have dealt with all suggestions in an adequate manner. We provide page and line numbers from both the submitted manuscript, as referenced by the referees, and from the revised manuscript with track changes attached at the end this document.

**Major comment 1 of Anonymous Referee #1**

In their paper, A. Cauquoin and C. Risi apply two different advection schemes (one of which in three different versions) in a GCM to test the influence of the representation of the advection on temperature and water isotope ratios in Antarctic precipitation. The paper is well structured and provides some interesting and important points on the uncertainties of Antarctic climate reconstruction that stems from model advection. In part, the paper lacks depth in method and interpretation of the results. The main conclusion that a physically more advanced advection scheme leads to more realistic results than a relatively simple one is not striking, however, the analysis is still a worthy endeavour. Before publication in GMD, I suggest to consider two major points, one concerning the method and one concerning the discussion, and a couple of minor points that are listed below.

- The paper only compares two advection schemes, one of which is (as described on page 3) obviously physically not as advanced as the other one. Hence, as already stated above, it is not surprising that the one captures the observations better. Some points can still be gained through this comparison, but it is a little disappointing that there are no further comparisons made. I am not aware of how complicated it is to run LMDZ with other advection schemes, but the study could go a lot further through that (and apparently this had even already been done in Cauquoin et al (2016), but the conclusions of that study are not discussed here). At the end it could be discussed what sort of GCM advection scheme ((semi-)Lagrangian, finite differences, explicit Eulerian,...) can be recommended to represent these processes well. The only, rather poor statement of the paper is that not too diffusive is better. Another form of enhancing the paper would be to try different horizontal resolutions. I assume that a higher horizontal resolution could, in particular for the "low-order" advection scheme, make a big impact in representing isotope ratios and temperature. Werner et al. 2011 show some study on this in Antarctica, here this issue is not even discussed (it only appears in the conclusion section without any connection to the rest of the paper). Some more work on this could easily enhance the (now pretty thin) message of the paper.

**Major comment of Anonymous Referee #2**

Cauquoin and Risi have investigated the effect of the advection scheme on the simulation of the water isotopes and temperatures over Antarctica. By conducting sensitivity tests with the LMDZ-iso model, they have concluded that the diffusivity of the advection scheme on the horizontal plane is crucial to simulate the water isotopes and temperatures over Antarctica. The paper is well written and structured. However, the scientific finding is very thin. The unacceptable level of the diffusivity property of the upstream scheme is a well-known fact and the sensitivity results are not surprising.

Therefore, I cannot recommend this manuscript for publication at this stage. As the authors argued, the diffusive property of AGCMs is attributable to the performance of the advection scheme, but it is at a much higher level than that of the upstream scheme. I believe that the upstream scheme is no longer used in current state-of-the-art AGCMs. Even the van Leer scheme is known as a diffusive scheme and using a less diffusive scheme is a challenging task. Actually, there are a number of more sophisticated advection schemes, but most of them are computationally expensive. Therefore, one may consider that the van Leer scheme is a reasonable choice in terms of accuracy and computational cost. I suggest that the authors should try more sophisticated schemes than the van Leer one and compare results in terms of accuracy and computational cost. Presumably, the computational cost is an important issue for paleoclimate studies, because they require very long-term simulations. Hourdin and Armengaud (1999) might be a good reference paper, in which not only the upstream scheme and van Leer schemes, but also other sophisticated schemes were tested based on the LMDZ model. If such advection scheme tests are done for the water isotopes and discussion is made from the paleoclimate view, that would be an attractive paper.

The only operational advection schemes in LMDZ with the water isotopes are the two schemes presented in this paper. As mentioned by the reviewer #2, Hourdin and Armengaud (1999) tested other schemes in LMDZ. However, it was with an old version of the model, and these other schemes have not been tested since a long time in LMDZ. For these two reasons, using other advection schemes with the LMDZ model was not possible within this study. Still, we follow the suggestion of the reviewer #1 and performed two additional simulations at a higher horizontal resolution (R144: latitude-longitude grid resolution of 1.27° × 2.5°) with the UP_xy and VL advection schemes. Increasing the horizontal resolution decreases both isotopic delta and temperature values, so in better agreement with the observations. So, refining the horizontal resolution plays the same role as improving the advection scheme on the horizontal plane. The best results are obtained when one improves both, horizontal resolution and advection scheme, at the same time. We added a table and paragraphs in the method and discussion sessions to highlight this point:

*(P4L1-4 – section 2) Increasing the grid resolution is equivalent to using an advection scheme that is less diffusive. Indeed, these finite-difference schemes are discretization methods and so depend on the chosen spatial resolution. To check that our findings and conclusions are consistent, we performed two more UP_xy and VL simulations but at the R144 resolution (latitude-longitude grid resolution of 1.27° × 2.5°).*

*(P7L14-23 – section 3.2.2) We test now the hypothesis that to increase the horizontal resolution is equivalent to using an advection scheme that is less diffusive. The Antarctica-mean results are summarized in the Table 2. Compared to the UP_xy R96 simulation, the average value of $\delta^{18}O$ in precipitation is decreased by 4.31‰ when the advection scheme is improved (VL R96), and by 2.42‰ when the horizontal resolution is increased (UP_xy R144), in better agreement with the observations. The picture is the same for the $\delta D$ outputs. The rather small decrease of the mean modeled temperature values is the same by changing the advection scheme or by increasing the resolution: by 0.85°C and 0.76°C respectively. The best results are reached by improving both the advection scheme and the horizontal resolution at the same time, with model-data differences in temperature and $\delta^{18}O$ of 4.83°C and 0.88‰ respectively. This confirms that an increase of the horizontal resolution plays the same role as an improvement in the representation of the advection scheme on its horizontal plane.*

**Major comment 2 of Anonymous Referee #1**

- The explanation for the conclusion that the temperature bias plays a minor role only for the isotope ratio bias is not fully convincing. Kinetic isotope fractionation is highly non-linear, in particular during desublimation of ice from the gas phase, which particularly happens at very low temperatures. A few degrees can have a large effect on the kinetic fractionation factor and thereby cause large differences in the isotope ratios. Fig. 2 also suggests a non-linear relationship between temperature and $\delta^{18}O$ below -40°. Your statement that the relations agree better above -40° strengthens the assumption that the temperature differences between model and observations are crucial here, also because the model does never simulate temperatures below -50°C. When you argue with the differences in RMSE here, you take this relation on to be purely linear, and that does not seem to hold for the entire range of data. Also, to my knowledge there are several microphysical isotope processes in the cloud and convection schemes not fully or not at all represented in the model water isotope physics. That should probably be discussed in this context, because it might have an impact at some temperature range. Since this is an important point for the conclusions of the paper, I suggest to add some more analysis and discussion on this topic to provide a convincing and more nuanced argumentation.

To answer to this point, we separated the discussion into two aspects and exploited more Figure 2:

1.  We made a first subsection 3.1 to study the model-data differences (from P4L18 to P6L12) in water stables isotopes (section 3.1.1), temperature (section 3.1.2) and spatial $\delta^{18}$O-temperature relationship (section 3.1.3). We point out that LMDZ has a warm bias and does not succeed to reach the coldest temperatures with the consequence that the model does not distillate enough. Furthermore, some kinetic and microphysical processes are represented using empirical relationships or not represented at all. All of this can contribute to an overestimation of the $\delta^{18}$O in precipitation (P5L14-17). For the non-linearity at very low $\delta^{18}$O/temperature, we do think that it is not related to first order to missing or poorly represented isotopic effects associated with kinetic fractionation. First, as for all GCM equipped with water isotopes, fractionation at sublimation is not taken into account in LMDZ-iso. This effect would however lead to further decrease of the water vapor $\delta^{18}$O in polar region and hence contribute to an even steeper $\delta^{18}$O-temperature slope at low temperature (hence further accentuate the non-linearity). A second effect that is not considered in LMDZ-iso is the Bergeron-Findeisen process. This process however occurs at intermediate temperature when both liquid and solid droplets coexist. It is thus not a good candidate to explain the non-linearity at low temperature. Finally, we agree that as in all the other models equipped with water isotopes, the parameterization of kinetic effect at vapor to solid condensation is represented empirically using a linear relationship between the supersaturation and the condensation temperature. A modification of the temperature can thus induce some change in the $\delta^{18}$O of the condensate. Using the formulation of Risi et al. (2010a) to define the supersaturation value and the equations from Jouzel and Merlivat (1984), a decrease in condensation temperature of 5°C (7.5°C decrease of surface temperature) for temperature characteristic of the cold central East Antarctic plateau has the effect to decrease the $\delta^{18}$O of the condensate by 2‰. Then, this effect of temperature on $\delta^{18}$O is of second order compared to the distillation effect explaining much of the 0.8‰.K$^{-1}$ slope between $\delta^{18}$O and surface temperature. Another possibility to illustrate the relatively modest effect of kinetic fractionation formulation on the $\delta^{18}$O-temperature slope is to draw the $\delta^{18}$O vs. temperature evolution resulting from distillation with and without kinetic effect linked to supersaturation. The figure below was realized using a simple water trajectory model equipped with isotopes and adapted to interpretation of $\delta^{18}$O and $\delta$D in polar regions (Mixed Cloud Isotopic Model, MCIM, Ciais and Jouzel (1994)). When the kinetic effect is switched off, the $\delta^{18}$O-temperature slope is lower than when the kinetic effect is switched on. As a consequence, with a model simulating a too high temperature over Antarctica (as for LMDZ-iso at low temperature), the kinetic effect is less important and the $\delta^{18}$O-temperature slope should decrease because of the change in kinetic effect. This is opposite to our non-linearity observation at very low temperature. To explain this steeper modeled

slope over the Antarctic plateau, we propose that, in the model, the water masses continue to be distillated when moving away from the coast, hence depleting the water vapor in heavy isotopes while the modeled temperature decrease from the coast to the remote region of the East Antarctic plateau is much less steep than in the reality (i.e. the model is not able to reproduce surface temperature below -50°C). This analysis is resumed in P6L1-12.

[Figure]

2. We made a second subsection 3.2 on the comparison between the different simulations. In the first paragraph of this subsection (P6L13-30), we analyze the differences in our modeled water isotope contents. We show that the very significant change in the initial $\delta^{18}O$ (more than 6‰) at -32°C can be attributed to the proportion of mixing against distillation that affects the water vapor during its transport (P6L26-30). In the second paragraph (P7L1-13), we focus on the temperature difference. We show a change in $\delta^{18}O$-temperature slope above -40°C, linked to Rayleigh distillation itself depending on the temperature. Because the change of temperature between the different simulations is rather small (~1°C between UP_xy and VL simulations), we expect a smaller impact of this temperature dependent formulation on our water isotope contents compared to the change of relative contribution between mixing and distillation (P7L10-14).

The conclusion has also been rewritten accordingly to this discussion (from P7L26 to P8L6).

**Minor comments of Anonymous Referee #1:**

- The abstract lacks to make the point about the impact of the temperature bias on the isotope ratio bias. I think this is a crucial point, but see also my major point.

See the answer to the major point.

(P1L15-17) *The temperature is also influenced, in a more minor way, by the diffusive properties of the advection scheme. A too diffusive horizontal advection increases the temperature and so also contributes to enrich the isotopic contents of water vapor over Antarctica through a reduction of the distillation.*

- P1 L11 and L13: "good" is a very relative term. This could also mean computationally efficient, and that would probably not match "your" meaning "good".

(P1L11) *... we show that a **less diffusive** representation of the advection, ...*

- P1L16: You know that these are isotop**ologues**. Please state this once and say that you call it isotopes (like in Werner et al., 2011). Is $H^{17}O$ included in your model? Why don't you evaluate it also in the study? Also, this sentence leaves open if you are talking about the "real world" or your model. In the real world, there are a couple more water isotopologues.

We mention now that these are isotopologues: (P1L19) *Water **stable isotopologues (hereafter designated by the term "water isotopes")**, are integrated tracer… (from P1L29 to P2L1)… water isotopes ($H_2^{16}O$, HDO, $H_2^{17}O$, $H_2^{18}O$) are now frequently…*

$H_2^{17}O$ is included in the model (Risi et al., 2013), but the lack of data and the limitations inherent to the GCMs make difficult to evaluate the spatio-temporal distribution of $^{17}O$-excess. We added the following sentence in the method section: (from P2L31 to P3L2) *LMDZ-iso is also able to simulate the $H_2^{17}O$ distribution (Risi et al., 2013) but we do not consider it here because the limitations inherent to the AGCMs lead to strong uncertainties and numerical errors on the spatio-temporal distribution of this isotope that make difficult its evaluation.*

- P1L18: Please explain very briefly what low accumulation sites are.

Done: (P1L21-22) *For example, low accumulation sites **that are typical on the East Antarctic Plateau (< 10 cm water-equivalent yr$^{-1}$)** provided the longest…*

- P1L20: Please name/list some of those "complex climate processes".

Done (P1L23-25): *of the numerous and complex **processes involved (water vapor transport, fractionation during the phase changes in the water cycle, distillation effect…)**.*

- P1L21: . This is particularly the case for the Antarctic, because this part of the world is subject to extreme weather conditions.

Done (P1L26-27)

- P2L1: Change "Indeed" to "For example"

Done (P2L5)

- P2L10-15: Here you talk only about studies that focus on vertical transport, although you said you would focus on horizontal advection, that is confusing. Is there no other work on horizontal advection with that respect? In the next paragraph you cite Cauquoin et al (2016), who compare two advection schemes. This seems to be relevant for the present paper, but you do not summarise their results in the context of the goal of the present study. Why not?

We modified this paragraph to clarify this point. We mention at the beginning of the paragraph that according to previous studies, the advection scheme has impacts on water isotopes on the horizontal plane as well as on the vertical direction. Then, we indicate that we test both.

(P2L11-13) … *much mixing.* ***According to previous studies, the diffusive properties of the advection scheme in the AGCMs, on the horizontal as well as on the vertical, can have an impact on the simulation of humidity and of its water isotope contents. On the horizontal,*** *dehydration* … (P2L16) *by steady advection (Hendricks et al., 2000).* ***On the vertical,*** *the excessive diffusion …*

(P2L23) … *to test if the warm and enriched biases in Antarctica are associated with an excessively diffusive water vapor transport,* ***both on the horizontal and on the vertical. Then, …***

- P2L32: "intrinsically more diffusive as explained below". This statement dangles in the nowhere here and confuses a little. You should either dedicate an entire sentence to the point or wait with it until it is explained below.

We removed this part of the sentence for less confusion (P3L7).

- P2L33: Remove "indeed"

Done (P3L8)

- P3L2-3: ...presented in the ...erroneously to the ... rather than to the van Leer scheme ...

Done (P3L10-11)

- P3L3: What results are that? And why do you nevertheless expect that fact to have a considerable influence on **your** results then?

These are the simulated isotopic results on a global scale. We corrected the sentence in this way (P3L12-14): …*and has little influence on their* ***simulated spatial and temporal distributions of water isotopes at a global scale. However, as we will show here, this has considerable effect on the spatial distribution of these proxies over region with extreme weather conditions such as Antarctica.***

- P3L5 For the sake...

Done (P3L16)

- P3L19: But also, it makes the upstream scheme physically not very sensible, right? Why would you use such an advection scheme anyway? Does it have advantages over the other scheme, or generally over other advection schemes? This is also part of my major point.

The advantage of this advection scheme is its cheaper cost in term of computing time. Moreover, it was historically the only scheme that worked with the water stable isotopes (Risi et al., 2010a). We applied this scheme with other spatial resolutions to test its impact on water isotope and temperature distribution over Antarctica. See our answer to the major comment of Anonymous Referees #1 and #2.

- P3L21: For reproducibility, can you say what SSTs you use? Is the simulation free running? Why this period? Is this the period where most data is available? Do you average the data over this period too?

These are AMIP simulations nudged by 20CR reanalyses (Compo et al., 2011): (P2L26-28) *We use here the isotopic AGCM LMDZ-iso (Risi et al., 2010a) following the model setup from Cauquoin et al. (2016)* **(AMIP (Gates, 1992) simulations forced by monthly observed sea-surface temperatures and nudged by the horizontal winds from 20CR reanalyses (Compo et al., 2011))** *at a standard latitude-longitude **R96** grid resolution (2.5° × 3.75°)...*

We have chosen this time period to do other studies by combining our simulated tritium outputs with the water stable isotopes. Moreover, such a period is long enough to have rather constant time-mean values. Masson-Delmotte et al. (2008) compiled all available observational Antarctic data sets (annual mean Antarctic surface temperatures, accumulation rates and present-day isotopic values) that are now broadly used by the modeling community (Risi et al. (2010a) or Werner et al. (2011) for example). These data are not necessarily over our time period but we do not expect any significant changes in our conclusions.

- P3L24-25: Can you estimate the uncertainty that bears the resolution of your model and this interpolation?

The difference in the RMSE values with and without interpolation, i.e. considering the nearest grid point, can be considered as an indicator of the effect of the resolution and interpolation on the model-data comparison, i.e. the effect of the **uncertainty associated with the model-data co-location**. Without interpolation, the RMSE of $\delta^{18}$O and temperature from the VL simulation differ by 0.5‰ (4.97‰) and 0.14°C (6.74°C). We mention this in P4L10-13: ***Without such an interpolation, i.e. considering the nearest grid point, the root-mean-squared errors (RMSE) of mean $\delta^{18}$O and temperature from the VL simulation differ only by 0.5‰ and 0.14°C compared to the results presented below, so the uncertainty associated with the model-data co-location is small.***

- P3L28: Why is this area "essential"?

We changed this sentence by (P4L15-16): *because **this area** provides the main reconstructions of past climate **based on the interpretation of water stable isotope records**.*

- P3L28: ...provides the main...climate and it ...

Done (P4L15-16), see answer to the previous comment.

- P3L29: Why? Do other regions in Antarctica not constitute an extreme test?

True, so we removed the last part *(it constitutes an extreme test for isotope-enabled AGCMs* (P4L16)) of this sentence.

- P4 and table 1: Why don't you include the results of the "xyz" simulation to the table as well? No need to include it to figure 1, but in the table it wouldn't harm.

Done

- P4 end of second paragraph: You should restructure this paragraph, this way it sounds odd. Suggestion: → Move the sentence about the poor representation of the boundary layer to the end of the paragraph. → Add another process that could possibly explain the rest of the bias → State that the investigation of these processes wrt the bias lies beyond the scope of this study.

Done (P5L8-14): *… However, the **average temperatures** of **-29.56°C and -31.54°C from** the **UP_xyz and VL simulations compared to the mean observed temperature (-36.93°C) shows that it corrects only in a marginal way the Antarctic warm bias in LMDZ-iso.** It has been suggested that the Antarctic warm bias in AGCMs could be linked to the general poor representation of the polar atmospheric boundary layer and related atmospheric inversion temperatures in these models (Krinner et al., 1997). **An underestimation of the cloud cover over the continent (Cesana and Chepfer, 2012) can also partly explain the overestimation of temperatures in Antarctica. The investigation of these processes with respect to the Antarctic warm bias lies beyond the scope of this study.***

- P5L15: and elsewhere too: horizontal plane

Done everywhere in the text

- P5L17: You do not really mean "globally", right? In all Antarctica?

We mean the average modeled temperature over Antarctica. (P5L22-23): *As noticed previously, the **average** modeled temperature **over Antarctica** is overestimated whatever the simulation considered.*

- P5L11: The aspect about higher resolution was never discussed, now it pops up in the conclusions, that should not be. Anyway, this is also a part of my major point.

We present now sensitivity tests to the horizontal grid resolution in the sections 2 (Model, simulations and method (P4L1-4)) and 3 (Results and discussion (P7L14-23)). See the answer to the major comment 1. We also changed the end of the conclusion accordingly (P8L9-10): ... *Another way to improve this aspect is to increase the spatial resolution, which has the same effect as applying a less diffusive advection scheme on the water isotopic composition and the temperature.*

**Minor comments of Anonymous Referee #2:**

P1. L13: "good" is ambiguous.
We changed *good* by *less diffusive* (P1L11).

P3, L1-L3: I cannot understand what this sentence means.
See answer to reviewer #1 (P3L10-14): *The advection scheme in the simulations presented **in** the LMDZ-iso reference paper from Risi et al. (2010a) was set erroneously **to the** simple upstream scheme rather than **to the** van Leer's scheme (Risi et al., 2010b), and has little influence on their simulated spatial and temporal distributions of water isotopes at a global scale. However, as we will show here, this has considerable effect on the spatial distribution of these proxies over region with extreme weather conditions such as Antarctica.*

P5, L7-9: "The fact that the advection… is not through temperature" is not fully convincing.
We clarified the analysis of the different causes that could change the isotopic content of precipitation and its relationship with temperature in link with the change of the advection scheme. See answer to major comment 2 of reviewer #1.

P6, L2-3: "The excessive diffusion… but is not the main cause" should be removed from Conclusion, because "its main cause is beyond of the scope of the study" (P4, L28).
Done (P7L26).

**Importance of the advection scheme for the simulation of water isotopes over Antarctica by general circulation models: a case study with LMDZ-iso (LMDZ5a revision 1750)**

Alexandre Cauquoin[1], Camille Risi[2]

[1]Alfred Wegener Institute, Helmholtz Centre for Polar and Marine Sciences, Bremerhaven, Germany
[2]Laboratoire de Météorologie Dynamique/Institut Pierre-Simon Laplace (LMD/IPSL), CNRS, Sorbonne Universités, UPMC Univ Paris 06, Paris, France

*Correspondence to*: Alexandre Cauquoin (alexandre.cauquoin@awi.de)

**Abstract.** Atmospheric general circulation models (AGCMs) are known to have a warm and isotopically enriched bias over Antarctica. We test here the hypothesis that these biases are consequences of a too diffusive advection. Exploiting the LMDZ-iso model, we show that a less diffusive representation of the advection, especially on the horizontal, is very important to reduce the bias in the isotopic contents of precipitation above this area. The choice of an appropriate representation of the advection is thus essential when using GCMs for paleoclimate applications based on polar water isotopes. Too much diffusive mixing along the poleward transport leads to overestimated isotopic contents in water vapor because dehydration by mixing follows a more enriched path than dehydration by Rayleigh distillation. The temperature is also influenced, in a more minor way, by the diffusive properties of the advection scheme. A too diffusive horizontal advection increases the temperature and so also contributes to enrich the isotopic contents of water vapor over Antarctica through a reduction of the distillation.

**1 Introduction**

Water stable isotopologues (hereafter designated by the term "water isotopes"), are integrated tracers of the water cycle. Especially, the isotopic composition recorded in polar ice cores enabled the reconstruction of past temperature variations (Jouzel, 2013 and references therein). For example, low accumulation sites that are typical on the East Antarctic Plateau (< 10 cm water-equivalent yr$^{-1}$) provided the longest ice core records, allowing to reconstruct past climate over several glacial-interglacial cycles (Jouzel et al., 2007). However, the interpretation of isotope signals remains challenging because of the numerous and complex processes involved (water vapor transport, fractionation during the phase changes in the water cycle, distillation effect…). This is particularly the case for Antarctica, because this part of the world is subject to extreme weather conditions.

To improve our knowledge on the mechanisms controlling the water isotopes distribution, atmospheric general circulation models (AGCMs) enhanced by the capability to explicitly simulate the hydrological cycle of the water isotopes ($H_2^{16}O$, HDO,

$H_2^{17}O$, $H_2^{18}O$) are now frequently used (Joussaume et al., 1984; Risi et al., 2010a; Werner et al., 2011). Water isotopes in climate models have been used, for example, to better understand how the climatic signal is recorded by isotopes in polar ice cores at paleoclimatic time scales (Werner et al., 2001).

5   However, some issues remain concerning the simulation of the climate over the Antarctic continent by AGCMs. For example, they frequently present a warm bias over this area (Masson-Delmotte et al., 2006) and isotopic values in precipitation that are not depleted enough compared to observations (Lee et al., 2007; Risi et al., 2010a; Werner et al., 2011). This raises the question why many of the AGCMs have these warm and enriched in heavy water isotopes biases over Antarctica.

10   In this paper, we hypothesize that these biases are associated with an excessively diffusive water vapor transport, i.e. transport that is associated with too much mixing. According to previous studies, the diffusive properties of the advection scheme in the AGCMs, on the horizontal as well on the vertical, can have an impact on the simulation of humidity and of its water isotope contents. On the horizontal, dehydration of air masses by mixing with a drier air mass leads to more enriched water vapor than dehydration by condensation and associated Rayleigh distillation (Galewsky and Hurley, 2010). For the same reason, poleward

15   water vapor transport by eddies (which act as mixing) leads to more enriched water vapor in Antarctica than transport by steady advection (Hendricks et al., 2000). On the vertical, the excessive diffusion during water vapor transport seems to be the cause of the moist bias found in most AGCMs in the tropical and subtropical mid and upper troposphere, and of the poor simulation of isotopic seasonality in the subtropics (Risi et al., 2012). The diffusivity of the advection scheme in the vertical has also important consequences on modeling of tracers like tritium, by affecting greatly its residence time in the stratosphere,

20   and so its downward transport from the stratosphere to the troposphere (Cauquoin et al., 2016).

The goal of this paper is to test if the warm and enriched biases in Antarctica are associated with an excessively diffusive water vapor transport, both on the horizontal and on the vertical. Then, we also test if the diffusive properties of the water vapor transport can be influenced by the prescribed resolution of an isotopic simulation.

25   **2 Model, simulations and method**

We use here the isotopic AGCM LMDZ-iso (Risi et al., 2010a) following the model setup from Cauquoin et al. (2016) (AMIP (Gates, 1992) simulations forced by monthly observed sea-surface temperatures and nudged by the horizontal winds from 20CR reanalyses (Compo et al., 2011)) at a standard latitude-longitude R96 grid resolution (2.5° × 3.75°), and with 39 layers in the vertical spread in a way to ensure a realistic description of the stratosphere and of the Brewer-Dobson circulation (Lott

30   et al., 2005). The model has been validated in polar regions for the simulation of both atmospheric (Hourdin et al., 2006) and isotopic (Risi et al., 2010a) variables. LMDZ-iso is also able to simulate the $H_2^{17}O$ distribution (Risi et al., 2013) but we do

not consider it here because the limitations inherent to the AGCMs lead to strong uncertainties and numerical errors on the spatio-temporal distribution of this isotope that make difficult its evaluation.

To quantify the effects of the prescribed advection scheme on water stable isotope values over Antarctica, we performed four
5   sensitivity simulations: (1) one control simulation with the van Leer (1977) advection scheme (called VL), which is a second order monotonic finite volume scheme prescribed in the standard version of the model (Risi et al., 2010a); and three other simulations whose the van Leer advection scheme has been replaced by a single upstream scheme (Godunov, 1959) on (2) the horizontal plane (UP_xy), (3) the vertical direction (UP_z), and (4) every directions (UP_xyz). Depending on one tunable parameter, the LMDZ model can be used with these 2 versions of the advection scheme according to the object of study (Risi
10   et al., 2012). The advection scheme in the simulations presented in the LMDZ-iso reference paper from Risi et al. (2010a) was set erroneously to the simple upstream scheme rather than to the van Leer's scheme (Risi et al., 2010b), and has little influence on their simulated spatial and temporal distributions of water isotopes at a global scale. However, as we will show here, this has considerable effect on the spatial distribution of these proxies over region with extreme weather conditions such as Antarctica.

For the sake of simplicity, we consider the advection along one dimension only, with wind flowing from grid box $i$-1 to grid box $i$ and from grid box $i$ to grid box $i$+1. In both Van Leer's (1977) second-order advection scheme and the upstream advection scheme (Godunov, 1959), the mixing ratio after advection in box $i$ ($q_i'$) is given by:

$$q_i' = \frac{q_i \times m_i + U_{i-1/2} \times q_{i-1/2} - U_{i+1/2} \times q_{i+1/2}}{m_i + U_{i-1/2} - U_{i+1/2}}, \tag{1}$$

20   where $q_i$ and $m_i$ are the mixing ratio and air mass in box $i$, $U_{i-1/2}$ is the air mass flux at the boundary between boxes $i$ and $i$-1, $U_{i+1/2}$ is the air mass flux at the boundary between boxes $i$ and $i$+1. The two schemes differ in the way the water vapor mixing ratio that is advected from box $i$-1 to $i$, $q_{i-1/2}$, and the water vapor mixing ratio that is advected from box $i$ to $i$+1, $q_{i+1/2}$, are calculated.

In Van Leer's scheme, $q_{i-1/2}$ is a linear combination of the mixing ratio in the boxes $i$-1 and $i$. Similarly, $q_{i+1/2}$ is a linear
25   combination of the mixing ratio in the boxes $i$ and $i$+1. For example, if the air mass flux from grid box $i$-1 to grid box $i$ is very small, then $q_{i-1/2} = (q_i + q_{i-1})/2$. This reflects the air that is advected into box $i$ is restricted to a small margin along the $i$-1/$i$ boundary, so its mixing ratio is exactly intermediate between $q_{i-1}$ and $q_i$.

In contrast, the upstream scheme is much simpler: $q_{i-1/2} = q_{i-1}$ and $q_{i+1/2} = q_i$. This means that even if the air mass flux from grid box $i$-1 to grid box $i$ is very small, the air that is advected into box $i$ has the same water vapor mixing ratio as grid box $i$-1 as a
30   whole. This makes the upstream scheme much more diffusive.

Increasing the grid resolution is equivalent to using an advection scheme that is less diffusive. Indeed, these finite-difference schemes are discretization methods and so depend on the chosen spatial resolution. To check that our findings and conclusions are consistent, we performed two more UP_xy and VL simulations but at the R144 resolution (latitude-longitude grid resolution of 1.27° × 2.5°).

We use here the LMDZ-iso outputs for the period 1990-2008. We express the isotopic composition of difference water bodies in the usual δ-notation as the deviation from the Vienna Standard Mean Ocean Water (V-SMOW). So for $H_2^{18}O$, the $\delta^{18}O$ value is calculated as $\delta^{18}O = ([H_2^{18}O]/[H_2^{16}O])_{sample} / ([H_2^{18}O]/[H_2^{16}O])_{V\text{-}SMOW} - 1) \times 1000$. Long-time mean δ values are then calculated as precipitation-weighted mean. For the quantitative model-data comparisons, we interpolate the LMDZ-iso

10 simulation outputs from the model grid to the same geographical coordinates as the observational datasets. Without such an interpolation, i.e. considering the nearest grid point, the root-mean-squared errors (RMSE) of mean $\delta^{18}O$ and temperature from the VL simulation differ only by 0.5‰ and 0.14°C compared to the results presented below, so the uncertainty associated with the model-data co-location is small. We make use of the observational database compiled by Masson-Delmotte et al. (2008) for analyzing the model performance over Antarctica. We also focus especially on the East-Antarctic plateau (defined by the

15 black bold contour of 2500 m above sea level elevation in Figure 1) because this area provides the main reconstructions of past climate based on the interpretation of water stable isotope records.

**3 Results and discussion**

**3.1 Model-data comparison**

**3.1.1 Water stable isotopes**

20 Figure 1 shows the observed annual mean $\delta^{18}O$ values in the snow surface in Antarctica compiled by Masson-Delmotte et al. (2008) (Figure 1a) and the difference with the modeled annual $\delta^{18}O$ in precipitation from the UP_z (Figure 1b), UP_xy (Figure 1c) and VL (Figure 1d) simulations. Our simulated $\delta^{18}O$ in precipitation is very sensitive to the choice of the advection scheme, with more enriched values when a more diffusive advection scheme is applied. The results from the VL simulation are in better agreement with the $\delta^{18}O$ observations over Antarctica (Figure 1d). This is confirmed by the root-mean-squared errors of

25 simulated $\delta^{18}O$ in precipitation from the UP_xyz and VL simulations, calculated as the difference between the observed annual mean values and the LMDZ-iso results, which are of 7.97‰ and 4.47‰ respectively (i.e. 21.7% and 12.2% of the observed mean Antarctic $\delta^{18}O$ value). The results from the VL simulation for the other isotopic variable δD is also the closest of the observations with RMSE of 40.93‰ (Table 1, red background). According to the observations, the East-Antarctic plateau is where the water isotope values are the lowest (mean $\delta^{18}O$ below -40‰, Figure 1a) due to the very low temperatures taking

30 place. Because of the extreme climate conditions at this area, one can see that the main disagreements between model outputs and observations are located at this place (Figure 1 and blue background of Table 1). Again, the isotopic outputs from the VL

simulation are in better agreement with the observations (Table 1, blue background). These first results confirm that an excessively diffusive water vapor transport influences significantly the simulated isotopic and temperature values over Antarctica.

**3.1.2 Temperature**

5 The bias in temperature is deteriorated about in the same way when applying a more diffusive advection on the vertical direction or on the horizontal plane, as shown with the RMSE of annual mean temperature of 7.50, 7.31 and 6.60°C for the UP_z, UP_xy and VL simulations respectively (Table 1, red background). This tendency is the same when focusing on the East-Antarctic plateau. However, the average temperatures of -29.56°C and -31.54°C from the UP_xyz and VL simulations compared to the mean observed temperature (-36.93°C) shows that it corrects only in a marginal way the Antarctic warm bias

10 in LMDZ-iso. It has been suggested that the Antarctic warm bias in AGCMs could be linked to the general poor representation of the polar atmospheric boundary layer and related atmospheric inversion temperatures in these models (Krinner et al., 1997). An underestimation of the cloud cover over the continent (Cesana and Chepfer, 2012) can also partly explain the overestimation of temperatures in Antarctica. The investigation of these processes with respect to the Antarctic warm bias lies beyond the scope of this study. This bias in LMDZ-iso, due mainly to the fact that the model does not succeed to reach the observed

15 coldest temperatures (see Figure 2), has the consequence that the distillation is not strong enough. Some microphysical processes and kinetic fractionation at very low temperature can be missed too. These different aspects could contribute to an overestimation of the $\delta^{18}O$ and $\delta D$ in precipitation over Antarctica.

**3.1.3 Spatial $\delta^{18}O$–temperature relationship**

We compare now our simulated spatial $\delta^{18}O$–temperature relationship and $\delta^{18}O$ values for a given temperature to the ones

20 from the data compiled by Masson-Delmotte et al. (2008). The spatial gradient for all the dataset is at 0.83‰/°C according to the VL simulation, very close of the observed one (0.80‰/°C). We make the same comparison but by restricting the dataset to the ones on the East-Antarctic plateau (Figure 2). As noticed previously, the average modeled temperature over Antarctica is overestimated whatever the simulation considered. Especially, no simulated temperature reaches a value below -50°C, and our $\delta^{18}O$ values are too depleted for the lowest simulated temperature range between -49°C and -45°C. The good agreement

25 between the simulated isotopic values in precipitation over the East-Antarctic plateau (VL, Figure 1d) and the observations despite too warm simulated temperatures could be explained by a bad representation of the atmospheric boundary layer and of its related inversion temperature. This would give a too warm surface temperature even if the condensation temperature in the model was close of the reality. A steeper modeled $\delta^{18}O$–temperature gradient is then observed for the lowest temperatures; if we restrict the fit to the modeled VL temperatures over -40°C, corresponding to the change of slope in our modeled $\delta^{18}O$–

30 temperature relation (thick orange line), the obtained simulated gradient (0.83‰/°C) is in very good agreement with the one from the observations (0.85‰/°C).

**Déplacé vers le bas [1]:** This is even more significant when focusing on the East-Antarctic plateau, with a model-data difference in $\delta^{18}O$ reaching 20‰ at some locations.

**Déplacé vers le bas [2]:** This lends support to our hypothesis that too much diffusive mixing along the poleward transport leads to overestimated $\delta^{18}O$ because dehydration by mixing follows a more enriched path than dehydration by Rayleigh distillation (Hendricks et al., 2000; Galewsky and Hurley, 2010).

The non-linearity of the modeled $\delta^{18}$O–temperature relationship at very low temperatures is unlikely related to first order to missing or poorly represented kinetic fractionation. First, as for all AGCMs equipped with water isotopes, fractionation at sublimation is not taken into account in LMDZ-iso. This effect would however lead to further decrease of the water vapor $\delta^{18}$O

5  in polar region and hence contribute to an even steeper $\delta^{18}$O-temperature slope at low temperature (hence further accentuate the non-linearity). As in all the other models equipped with water isotopes, the parameterization of kinetic effect at vapor to solid condensation is represented empirically using a linear relationship between the supersaturation and the condensation temperature (Risi et al., 2010a). A modification of the temperature can thus induce some change in the $\delta^{18}$O of the condensate but cannot explain the data-model mismatch. To explain this steeper modeled slope over the Antarctic plateau, we propose

10  that, in the model, the water masses continue to be distillated when moving away from the coast, hence depleting the water vapor in heavy isotopes while the modeled temperature decrease from the coast to the remote region of the East Antarctic plateau is much less steep than in the reality (i.e. the model is not able to reproduce surface temperature below -50°C).

**3.2 Comparison of the different simulations**

**3.2.1 Effects of the diffusive properties of the advection scheme**

15  We compare here the results from our different simulations at a R96 grid resolution. The UP_z simulation (upstream vertical advection, Figure 1b) increases a little the bias in $\delta^{18}$O, but its results stay relatively close of the $\delta^{18}$O values from the VL simulation, indicated by the similar average values that differ only by 0.89‰ for all Antarctica (Table 1). On the other hand, the $\delta^{18}$O outputs from the UP_xy simulation (upstream horizontal advection, Figure 1c) display greater differences with the VL simulation ones, and so with the isotopic data, as revealed by the mean UP_xy - VL difference in $\delta^{18}$O of 4.31‰. This is

20  even more significant when focusing on the East-Antarctic plateau, with a model-data difference in $\delta^{18}$O reaching 20‰ at some locations. The annual mean $\delta^{18}$O and $\delta$D values from the UP_xy simulation are increased by 6.39‰ and 44.26‰ compared to the VL simulation average values, in less agreement with the observations as shown by their respective RMSE values (Table 1, blue background). It shows that the diffusive property of the advection scheme on the horizontal plane is essential to better model the water isotope distribution, especially over Antarctica. To go further, one can also compare the

25  $\delta^{18}$O values at a fixed temperature for the UP_xy and VL simulations (Figure 2, red and orange crosses respectively). The $\delta^{18}$O in precipitation for a temperature of -32°C over the East-Antarctic plateau is already smaller by 6.1‰ in the VL simulation. This very significant difference in initial $\delta^{18}$O can be attributed to the proportion of mixing against distillation that affects the water vapor during its transport. This lends support to our hypothesis that too much diffusive mixing along the poleward transport leads to overestimated $\delta^{18}$O because dehydration by mixing follows a more enriched path than dehydration by

30  Rayleigh distillation (Hendricks et al., 2000; Galewsky and Hurley, 2010).

Déplacé (insertion) [1]

Déplacé (insertion) [2]

As noticed in section 3.1, all our simulations overestimate the average temperature in Antarctica and even more on the East-Antarctic plateau. A more diffusive advection on the horizontal or on the vertical increases the mean temperature value by 0.85 and 1.03°C respectively compared to the VL result. To explain such an influence of the advection on the temperature over Antarctica, even secondary, one can hypothesize that the Antarctic continent is better isolated, and so colder, when the

5 advection of the model is less diffusive. If we focus now on the link between the temperature and the $\delta^{18}$O in precipitation, the $\delta^{18}$O-temperature gradients according to our different R96 simulations UP_z, UP_xy and VL are at 0.79, 0.69 and 0.83‰/°C respectively. The difference between the VL and UP_xy gradient shows an effect of diffusive properties of the large-scale transport on the distillation process, this later depending on temperature. Such a difference between the modeled $\delta^{18}$O-temperature gradients is also observed if we restrict to the temperature range above -40°C, with gradients of 0.95 and 0.83‰/°C

10 according to the UP_xy and VL simulations respectively (Figure 2, red and orange thick lines). As the modeled temperature difference between the UP_xy and VL simulation is small, i.e. less than 1°C, we do not expect such a strong effect of temperature change on water isotope content in precipitation through the change of slope of the Rayleigh distillation compared to the change of the relative contribution between mixing and distillation.

**3.2.2 Effects of the horizontal grid resolution**

15 We test now the hypothesis that to increase the horizontal resolution is equivalent to using an advection scheme that is less diffusive. The Antarctica-mean results are summarized in the Table 2. Compared to the UP_xy R96 simulation, the average value of $\delta^{18}$O in precipitation is decreased by 4.31‰ when the advection scheme is improved (VL R96), and by 2.42‰ when the horizontal resolution is increased (UP_xy R144), in better agreement with the observations. The picture is the same for the $\delta$D outputs. The rather small decrease of the mean modeled temperature values is the same by changing the advection scheme

20 or by increasing the resolution: by 0.85°C and 0.76°C respectively. The best results are reached by improving both the advection scheme and the horizontal resolution at the same time, with model-data differences in temperature and $\delta^{18}$O of 4.83°C and 0.88‰ respectively. This confirms that an increase of the horizontal resolution plays the same role as an improvement in the representation of the advection scheme on its horizontal plane.

**4 Conclusions**

25 We have tested with LMDZ-iso if the warm and isotopically enriched biases in Antarctica, frequently observed in the AGCMs, are associated with the diffusive property of the advection scheme. The simulated water isotope contents in Antarctica are very sensitive to the diffusive character of the water vapor transport, especially on the horizontal plane. The higher the contribution of mixing (i.e. diffusion) is important, the more enriched the precipitation. These findings are even more striking for the East-Antarctic plateau where the main ice cores allowing paleoclimate reconstructions are located. Moreover, because the diffusive

30 character of the large-scale transport influences the temperature in this region, even in a light way, this has an impact on the modeled water isotopic composition through the Rayleigh distillation. So, we conclude here that the excessive numerical

diffusion has a large influence on the enriched isotopic bias. For the isotope–temperature relationship over Antarctica observed in LMDZ-iso, this latter is improved for the temperatures above -40°C. At the lowest temperatures (i.e over the East-Antarctic plateau), that the model is not able to reach, the non-linearity observed in our simulations can be unlikely explained at first order to missing or poorly represented kinetic fractionation. One can speculate that the water masses continue to be distillated when moving away from the coast, hence depleting the water vapor in heavy isotopes while the modeled temperature decrease from the coast to the remote region of the East Antarctic plateau is much less steep than in the reality. Our study demonstrates that a representation of the advection scheme in the AGCMs taking into account water isotopes and isotopic gradients, especially on the horizontal domain, is an important step toward a more realistic modeling of water isotopes over Antarctica. Another way to improve this aspect is to increase the spatial resolution, which has the same effect as applying a less diffusive advection scheme on the water isotopic composition and the temperature.

**5 Code availability**

The code of the LMDZ5a model (revision 1750 without water isotopes), on which LMDZ-iso is based on, can be downloaded via the command: svn checkout http://svn.lmd.jussieu.fr/LMDZ/LMDZ5/branches/testing@1750 LMDZ5. General information about the model and its documentation can be found on http://lmdz.lmd.jussieu.fr and on http://lmdz.lmd.jussieu.fr/utilisateurs/manuel-de-reference-1/lmdz5-documentation/view respectively. The LMDZ-iso code is available as a supplement of this manuscript.

*Acknowledgements*. We thank A. Landais for her useful suggestions on this manuscript. LMDZ simulations have been performed on the Ada machine at the IDRIS computing center under the GENCI project 0292. The research leading to these results has received funding from the European Research Council under the European Union's Seventh Framework Programme (FP7/20072013)/ERC grant agreement no. 30604.

[revised manuscript text omitted]

Cellules insérées

Cellules insérées

[Figure]

**Figure 2:** Relationship between $\delta^{18}$O and temperature on the East-Antarctic plateau according to the observations (black) and the UP_z (blue), UP_xy (red) and VL (orange) simulations. For each simulation outputs, two linear regressions have been conducted: one on the full East-Antarctic plateau dataset (thin lines) and one on the same dataset without the temperatures below -40°C (bold lines). The corresponding formulas of these latters are also shown.

| | Mean data | Mean UP_xy R96 | Mean VL R96 | Mean UP_xy R144 | Mean VL R114 |
|---|---|---|---|---|---|
| T (°C) | **-36.93** | -30.69 | -31.54 | -31.45 | -32.10 |
| $\delta^{18}O$ (‰) | **-36.76** | -31.43 | -35.74 | -33.85 | -37.64 |
| $\delta D$ (‰) | **-289.62** | -251.34 | -279.49 | -267.17 | -289.55 |

**Table 2: Comparison of the observed annual mean values of temperature (T), $\delta^{18}O$ and $\delta D$ for the full Antarctic dataset with four different LMDZ-iso simulations, combining different horizontal resolutions (R96 and R144) and different advection schemes (UP_xy and VL).**